# Coronary Artery Disease Detection Based on a Novel Multi-Modal Deep-Coding Method Using ECG and PCG Signals

**DOI:** 10.3390/s24216939

**Published:** 2024-10-29

**Authors:** Chengfa Sun, Changchun Liu, Xinpei Wang, Yuanyuan Liu, Shilong Zhao

**Affiliations:** Department of Biomedical Engineering, School of Control Science and Engineering, Shandong University, Jinan 250061, China; 201920518@mail.sdu.edu.cn (C.S.); liuyy@sdu.edu.cn (Y.L.); zhaoshilong@mai.sdu.edu.cn (S.Z.)

**Keywords:** CAD, multi-modal method, recurrence plot, deep learning, feature extraction

## Abstract

Coronary artery disease (CAD) is an irreversible and fatal disease. It necessitates timely and precise diagnosis to slow CAD progression. Electrocardiogram (ECG) and phonocardiogram (PCG), conveying abundant disease-related information, are prevalent clinical techniques for early CAD diagnosis. Nevertheless, most previous methods have relied on single-modal data, restricting their diagnosis precision due to suffering from information shortages. To address this issue and capture adequate information, the development of a multi-modal method becomes imperative. In this study, a novel multi-modal learning method is proposed to integrate both ECG and PCG for CAD detection. Along with deconvolution operation, a novel ECG-PCG coupling signal is evaluated initially to enrich the diagnosis information. After constructing a modified recurrence plot, we build a parallel CNN network to encode multi-modal information, involving ECG, PCG and ECG-PCG coupling deep-coding features. To remove irrelevant information while preserving discriminative features, we add an autoencoder network to compress feature dimension. Final CAD classification is conducted by combining support vector machine and optimal multi-modal features. The experiment is validated on 199 simultaneously recorded ECG and PCG signals from non-CAD and CAD subjects, and achieves high performance with accuracy, sensitivity, specificity and f1-score of 98.49%, 98.57%,98.57% and 98.89%, respectively. The result demonstrates the superiority of the proposed multi-modal method in overcoming information shortages of single-modal signals and outperforming existing models in CAD detection. This study highlights the potential of multi-modal deep-coding information, and offers a wider insight to enhance CAD diagnosis.

## 1. Introduction

Coronary artery disease (CAD), a prevalent cardiovascular disorder, contributes significantly to a high mortality rate globally [1]. It arises from atherosclerosis, characterized by extensively accumulating cholesterol and fatty plaques along coronary walls. These plaques cause various physical ailments, involving fatigue, dizziness, myocardial ischemia, and even myocardial infarction in severe cases [2]. Terminal CAD progresses irreversibly, and eventually leads to the death of the patient. Therefore, timely and precise detection is essential in slowing CAD progression and improving the survival rate of patients [3]. Coronary angiography is the gold standard for diagnosing CAD, but its invasive and high-cost nature constrains its widespread use. In contrast, electrocardiogram (ECG) and phonocardiogram (PCG) are both noninvasive and cost-friendly tools for early CAD screening. However, due to the complexity of cardiovascular activities, single-modal data contain limited identifying information and present information shortages in CAD analysis. Multi-modal learning techniques by integrating diverse sources of information can mitigate this shortcoming and provide more adequate information for better revealing cardiovascular conditions. Consequently, multi-modal information can achieve better classification performance and attract more attention in diverse fields.

ECG, the most commonly used clinical tool, records microstructural changes of cardiac electrical signal during each heartbeat. However, early and moderate CAD patients manifest negligible symptoms in ECG waves, so these patients may be incorrectly diagnosed [4]. Conversely, PCG records synthesized sounds of heart mechanical vibration from autonomous movement of cardiac tissues, involving myocardial contraction, valve vibration and blood flow striking against artery walls. When one or more coronary arteries are occluded, blood flows through the narrowed vessels and forms turbulence, producing high-frequency murmurs. Nevertheless, in CAD cases with blockages more than 95%, weak high-frequency heart murmurs from turbulence in almost blocked coronary arteries may diminish or disappear; thus, it is difficult to observe weak diastolic murmurs from these patients’ PCG signals, and this poses a challenge in CAD detection [5,6]. As mentioned by the above studies, only single ECG or single PCG provides insufficient information in CAD diagnosis. Multi-modal information, combing both ECG and PCG information, can help physicians arrive at a more objective and accurate diagnosis. Moreover, ECG and PCG waves exhibit close temporal correlations during each corresponding heartbeat, such as the onset of the first heart sound closely aligning with the R-wave peak of the ECG, and the second heart sound being located close to the termination of the T-wave [7]. This synchronized nature underscores the potential of multi-modal learning methods that integrate both ECG and PCG signals and offer a wider view to better reveal cardiovascular conditions.

Numerous automatic detection techniques based on single ECG [8,9,10] or single PCG [11,12,13,14] for CAD classification have been widely proposed. However, multi-modal methods for analyzing CAD remain relatively scarce. Recently, the correlation between ECG and PCG has been addressed. An improved D-S theory fused ECG and PCG signals for cardiovascular disease identification and achieved superior accuracy over single ECG or single PCG by utilizing wavelet scattering transform to extract multi-modal time–frequency features [15]. Zarrabi et al. [16] proposed a novel decision system based on multi-modal features combining ECG, PCG and clinical data to predict the risk of myocardial infarction, and outperformed single-modal features. Li et al. [17] designed a novel dual-input neural network integrating ECG with PCG for classifying non-CAD and CAD, and the final testing result indicated that dual-input data had superior performance over single-input data. In the latest study, Li et al. [7] further validated the advantage of multi-modal learning by constructing ECG-net and PCG-net for encoding deep features from ECG and PCG, confirming the superiority of their multi-model method. Additionally, electromechanical coupling information between time intervals of ECG and PCG has been proposed [18]. Dong et al. [19] utilized a novel coupling analysis method based on time intervals of ECG and PCG signals to assess coronary blockage degree, and several types of entropies and cross-entropies were utilized to analyze coronary blockage degree, which underscores the potential of multi-modal fusion in providing more accurate information of the cardiovascular condition.

In the process of computer-aided CAD detection, feature extraction is key in characterizing ECG and PCG signals. A variety of linear features, including time-domain [11,20,21,22], frequency-domain [11,20,21,22] and time–frequency features [8,13,20], have been presented to represent disease-related information. Moreover, considering the inherent nonlinear nature of ECG and PCG signals, various nonlinear features, encompassing multiple types of entropies [10,19], recurrence plot [23,24,25] and multi-fractal parameters [26,27], were also developed for anomaly classification. More recently, deep learning methods with complex structural layers were used to encode ECG and PCG deep features for detecting CAD [9,12,14]. Notably, convolutional neural network (CNN) [28], serving as the most common feature extractor, automatically encodes signal-based or image-based deep features by operating various convolutional and pooling layers. Additionally, long short-term memory (LSTM) [29], bidirectional long short-term memory (BiLSTM) [30], transformer [31], autoencoder [32] and their improved models [33,34] were also adopted to process ECG and PCG, enhancing feature extraction and abnormality detection abilities. However, deep learning methods often yield high-dimensional features that inevitably contain some amount of redundant information, adversely influencing classification performance. To solve this problem, Principal Component Analysis (PCA) [20], a classic dimension reduction technique, was employed. Furthermore, the autoencoder network was also utilized for data compression and feature selection [35], offering an alternative approach to refine the feature space.

Apart from effective feature extraction, CAD classification performance also relies on efficient classifiers. A large number of classifiers have been proposed for CAD detection and prediction, involving support vector machine (SVM) [8,10,22,36], Naïve Bayes [36], decision tree (DT) [36,37], K-nearest neighbor (KNN) [38], boosting and bagging model [13], and artificial neural network (ANN) [9,10,11,12,20,21]. In addition, a novel classification decision strategy based on the integration of manual and automated thresholding techniques was employed to identify abnormal ECG [39].

According to the reference, this study proposed a novel multi-modal learning method to consider both ECG and PCG for CAD detection. We initially carried out the deconvolution of ECG and PCG and produced a novel ECG-PCG coupling signal to reveal the inherent relationship of ECG and PCG. Then ECG, PCG and ECG-PCG coupling signals were transformed into modified recurrence plots (MRPs) to quantify their respective nonlinear dynamic microstructural information. A parallel CNN model was constructed for encoding deep features from each MRP, and then we fused these single-modal deep-coding features into multi-modal information. To remove redundant information while preserving discriminative features, we added an autoencoder network behind the parallel CNN network to reduce feature dimension. A combination of optimal features and SVM classifier was used for final classification. The diagram of this proposed method is shown in Figure 1.

The highlights of this study are as follows:

A novel multi-modal learning method by considering both ECG and PCG signals is proposed to detect CAD.The multi-modal deep-coding information involves ECG, PCG and ECG-PCG coupling deep-coding MRP features.The proposed method constructs MRPs to quantify the nonlinear dynamic characteristics of ECG, PCG and their deconvolution signals, and builds the integrating deep learning network to code multi-modal deep-coding features and reduce feature dimension.A combination of optimal multi-modal features and SVM classifier is used for final classification, and the result indicates superiority of the multi-modal learning method.

The remaining sections are organized as follows. Section 2 describes data preprocessing, ECG-PCG coupling signal evaluation, MRP construction, deep learning network construction and performance evaluation. Section 3 illustrates experimental results, while Section 4 compares and discusses the performance of different models. The final conclusion is in Section 5.

## 2. Materials and Methods

### 2.1. Data

A total of 199 subjects, recruited at Qianfoshan Hospital (Shandong First Medical University Affiliated Hospital) in Jinan, Shandong Province, China, participated in this study. This experiment was conducted with the permission of the hospital Ethical Review Committee (ethics approval number: S374) and adhered strictly to the guidance of the Declaration of Helsinki and its amendments. All subjects presented various symptoms, involving chest tightness, chest pain, and palpitations over a week, and then signed informed consent before participation. This study excluded individuals who had undergone an intervention of percutaneous coronary and coronary artery bypass surgery, and had been diagnosed with valvular diseases. Each participant underwent coronary angiography and then their diagnosis results were decided by the professional physician based on the results of coronary angiography. Those with blockages ≥ 50% in at least one major coronary artery (left anterior descending, left circumflex, or right coronary artery) were diagnosed as CAD (135 positive cases); others were diagnosed as non-CAD (64 negative cases). Before conducting the experiment, the basic information of subjects is recorded in Table 1, including age, sex, height, weight, heart rate and blood pressure.

To ensure more precise resting data collection, each subject lay supine for at least 10 min in a quiet and controlled temperature (25 ± 3 °C) room. A cardiovascular function detector (CVFD-II, Huiyironggong Technology Co., Ltd., Jinan, China) was employed to simultaneously record standard lead-II ECG and PCG signals for 5 min at a sampling rate of 1000 Hz. As CAD mostly affects the left coronary artery, the electronic stethoscope is positioned in the third intercostal space on the left edge of the sternum for recording the PCG signal. The subjects remained calm and awake during the process of the experiment.

### 2.2. Data Pre-Processing

#### 2.2.1. Data Denoising

The original collected signals contain various types of noises. To obtain clean ECG and PCG signals, a 0.5–60 Hz Butterworth bandpass filter and a 20 Hz high-pass Butterworth filter were applied to process raw ECG and PCG signals, respectively. And then, an IIR notch filter was utilized to remove power frequency interference from both signals. After that, the pre-processed ECG and PCG signals were cropped into 10 s segments, which were regularized using the z-score normalization to further extract features. Figure 2a,b and Figure 3a,b show ECG and PCG segments from a non-CAD subject and a CAD patient, respectively. As shown in Figure 2a and Figure 3a, the CAD patient ST-waveform in the ECG presents a significant elevation compared with the non-CAD subject. Additionally, in the comparison with the PCG of the CAD patient, as shown in Figure 2b and Figure 3b, the non-CAD subject PCG has obvious boundaries around the first heart sound and second heart sound.

#### 2.2.2. ECG-PCG Coupling Signal Evaluation

ECG and PCG signals convey a great deal of information concerning cardiac electrical activity and mechanical activity, respectively. These two activities are intricately connected through electromechanical coupling and mechanical–electrical feedback mechanisms. Relying on ECG-PCG coupling analysis, we can acquire the intrinsic relationship between ECG and PCG for reflecting complex cardiovascular activities. It contains an amount of effective information concerning cardiovascular disease identification. Hence, we conduct deconvolution of ECG and PCG and produce a novel ECG-PCG coupling signal, further enhancing the understanding of cardiac function.

It should be noted that electrical activity of the heart occurs initially, and then it propels mechanical activity. Therefore, a novel electromechanical coupling system model is designed by using ECG as the input and PCG as the output, based on the sequence of cardiac activity, as defined in Equation (1).
(1)y(n)=x(n)∗h(n)

Here, * denotes convolution operation. *y*(*n*) denotes the PCG signal with 2*N* − 1 sample points. *x*(*n*) is the ECG signal with *N* sample points and *h*(*n*) is the novel ECG-PCG coupling signal with *N* sample points. Along with deconvolution calculation, the ECG-PCG coupling signal *h*(*n*) is evaluated successfully, and it shows nonlinear behavior due to the nonlinearity of ECG and PCG.

In the process of deconvolution operation, *x*(*n*) and *h*(*n*) are padded with *N* − 1 zeros until the lengths of both signals reach 2*N* − 1. This preparation facilitates the transformation of the one-dimensional input signal *x*(*n*) into the matrix X with rank-*L*, thereby converting the convolution operation in Equation (1) into a more efficient matrix-vector calculation format in Equation (2).
(2)y=Xh
where X is rank-*L* convolution matrices of the form
(3)$$X=\left[\begin{array}{ccccc} x(0) & x(L-1) & x(L-2) & \cdots & x(1)\\ x(1) & x(0) & x(L-1) & \cdots & x(2)\\ x(2) & x(1) & x(0) & \cdots & x(3)\\ \vdots & \vdots & \vdots & \vdots & \vdots \\ x(L-1) & x(L-2) & x(L-3) & \cdots & x(0) \end{array}\right]$$

Here, *L* is equal to 2*N* − 1. y is the column vector of output signal *y*(*n*) and h is the column vector containing electromechanical system parameters.
$$y={\begin{array}{ccccc} \left[y(0\right) & y(1) & y(2) & \cdots & y(2N-1)] \end{array}}^{T}$$
$$h={\begin{array}{ccccc} \left[h(0\right) & h(1) & h(2) & \cdots & h(2N-1)] \end{array}}^{T}$$
where (.)*^T^* represents vector transposition. Within the operating matrix calculation in Equation (2), *h*(*n*) is evaluated successfully. Figure 2c and Figure 3c vividly display the ECG-PCG coupling signals of a non-CAD subject and a CAD patient, respectively. Notably, there exist more significant differences between CAD and non-CAD subjects. In the ECG-PCG coupling signal, non-CAD subject has a less change in waves, whereas CAD patient waves change more obviously.

### 2.3. Modified Recurrence Plot

Given the inherent nonlinear and non-stationary nature of physiological signals, the nonlinear dynamic analysis can more accurately portray the characteristics of these signals. Consequently, we transform each one-dimensional signal into the modified recurrence plot (MPR) based on signal phase space reconstruction to observe hidden patterns and microstructural forms of cardiovascular status, thereby providing valuable insights into the underlying dynamics.

#### 2.3.1. Phase Space Reconstruction

The time series signal can be transformed into a vector form by reconstructing a phase space based on embedding theorem [40]. Specifically, given a time series signal *x*(*t*) with *N* sample points, it can be reconstructed into a new phase space comprising *N* − *m* + 1 vectors, expressed as:(4)X(i)=[x(i)x(i+τ)⋯x((i+(m−1)τ)],i=0,1,⋯,(N−(m−1))τ

Here, *X*(*i*) is a vector in phase space. *m* and *τ* are the embedding dimension and time delay, respectively. These critical learning parameters can be approximated using the false nearest neighbor algorithm [41] and mutual information algorithm [42]. Due to the influence of individual specificity, the values of *m* and *τ* vary significantly, and inappropriate values can ignore the amount of detail in the signal. Thus, the selection of the appropriate embedding dimension and time delay is essential. In this study, the difference between non-CAD and CAD subjects mainly manifests in detailed changes in physiological signals. To observe more details, we reconstruct phase spaces of ECG, PCG and ECG-PCG coupling signals with *τ* = 1 and *m* = 1 [43], and further construct their novel modified recurrence plots on this basis, which provides a more comprehensive understanding of the underlying dynamics and hidden patterns in the signal.

#### 2.3.2. Modified Recurrence Plot Construction

In accordance with phase space reconstruction, the nonlinear dynamic signal shows the significant recurrence characteristics [44]. Thus, this work attempts to further construct the novel recurrence plots for quantifying the microstructure of ECG, PCG and ECG-PCG coupling signals, respectively. The traditional recurrence plot (RP), an effective nonlinear signal processing method, maps the phase space vectors to the two-dimensional image form, thereby visualizing the dynamic change information of physiological signals, which is defined as Equation (5).
(5)$$R_{ij}=\Theta (\varepsilon -\left\|X_{i}-X_{j}\right\|)$$
(6)$$\Theta (x)=\left\{\begin{array}{l} 0,x<0\\ 1,x\geq 0 \end{array}\right.$$
where ε is the threshold value. ‖·‖ denotes Euclidean distance. *X*(*i*) and *X*(*j*) are phase space vectors. Θ () is the Heaviside function, as expressed in Equation (6).

The threshold *ε*, a pivotal parameter in traditional RP construction, determines the values of recurrence points in the form of a grayscale image. When *ε* exceeds the distance between any two vectors, the corresponding value of the recurrence plot is 1; otherwise, it is 0. The smaller threshold ensures adequate recurrence points, while the larger threshold may be necessitated in the presence of noise, as noise can distort the structure of traditional RP.

Despite the fact that traditional RP can offer insights into the nonlinear characteristics of cardiovascular status to a certain degree, the amount of detailed physiological changes in signal waves still fails to be observed. To overcome this limitation, a modified recurrence plot (MRP) is introduced, dispensing with the threshold value and instead utilizing color codes to characterize the distances between phase space vectors. It maps the distances between the vector at time *i* and all vectors into a color scale, as defined Equation (7), enabling a more nuanced visualization of signal dynamics.
(7)$$\upsilon \left(i,j\right)=\vartheta (\left\|\vec{X}(i)-\vec{X}(j)\right\|)$$

Here, ‖·‖ represents Euclidean distance, and υ represents the color code that maps the distance to the color scale. The color code assigned to the pair of vectors *X*(*i*) and *X*(*j*) is located at the coordinate (*i*, *j*) in the novel MRP, which quantifies more nonlinear information of the physiological signal. The MRP employs a gradient of darkness to signify closer distances between vectors, while brighter points represent farther distances. Figure 2 and Figure 3 illustrate MRPs of ECG, PCG and ECG-PCG coupling signals from non-CAD and CAD subjects, respectively. In comparison with the non-CAD subject, the MRPs of the CAD patient exhibit more notable alterations than those of non-CAD individuals.

### 2.4. Feature Extraction Based on Integrating Deep Learning Network

The deep learning method shows numerous advantages and it proves to be highly effective for feature extraction and anomaly classification in diverse fields. Particularly, CNN automatically encodes the spatial information of the image and confers significant benefits in image recognition tasks. To extract effective disease-related features from MRPs for identifying CAD, we built a model integrating deep learning by adding a parallel CNN and an autoencoder network to encode deep feature representations from each MRP, and the frame is shown in Figure 4. In the proposed model integrating deep learning, the parallel CNN network encoded deep features reflecting detailed information from different MRPs, and then fused three single-modal features to form multi-modal information. Meanwhile, the autoencoder network was utilized to compress feature dimensions and acquire more meaningful disease-related features.

#### 2.4.1. The Parallel CNN Network

In this study, a parallel CNN model is built to encode a deep feature representation from the constructed MRPs, leveraging advantages of the deep learning network. The parallel CNN model, comprising three independent CNN branches, processes ECG, PCG and ECG-PCG coupling MRPs for encoding different single-modal features. Then, the outputs of three CNN branches were concatenated to form multi-modal features.

MRPs of ECG, PCG and ECG-PCG coupling signals are initially resized to 224 × 224 × 3 as the input of the proposed parallel CNN model to encode deep features of each single-modal signal. Each CNN branch of the proposed parallel model contains 13 convolutional layers with 3 × 3 kernel, as shown in Figure 5. All convolutional layers are organized into 5 sections, which contain 64, 128, 256, 512, and 512 convolutional kernels, respectively. A maximum pooling layer with 2 × 2 kernel is located at the end of each section. All parameters of convolutional and pooling layers are detailed in Table 2. The activation function of all hidden layers is the Rectified Linear Unit (ReLU). In the proposed network integrating deep learning, convolutional and pooling layers are core modules for feature extraction. The output of each layer is termed as features carrying an amount of temporal and spatial information of the image. The quantity of spatial image features depends on the number of different kernels in each layer. It is defined as Equations (8) and (9).
(8)$$y_{j}=\sum_{i}z_{ij}\times x_{i}+b_{i}$$
(9)$$y_{t}=\max(\sum_{n}z_{nt}\times x_{n})$$
where *x_i_* and *x_n_* represent the input image features as inputs of convolutional and max pooling layers, respectively. *y_j_* and *y_t_* denote the outputs of convolutional and max pooling layers, respectively. *z_ij_* and *z_nt_* are convolutional and max pooling kernels, respectively. *b_i_* is the bias.

By operating the parallel CNN model, the feature images, the new output of the last pooling layer, are flattened into a high one-dimensional deep feature vector. Considering the over-fitting condition of the high-dimensional feature vector, we implemented a dimension reduction technique to enhance the generalization ability of the model.

#### 2.4.2. Autoencoder Network

This study utilized the parallel CNN network to encode different single-modal deep features from each signal, and then fused them to create high-dimension multi-modal features that inevitably included an amount of redundant information. To remove information redundancy of the high-dimensional features while preserving more discriminative features, an autoencoder network following behind the parallel CNN network was introduced to reduce feature dimension and preserve more salient features. The proposed autoencoder network includes an input, an output and five fully connected hidden layers, which consist of two parts: encoder and decoder, as shown in Figure 6. All parameters of the autoencoder network are listed in Table 3. The encoder maps the high-dimensional input x, which contains both useful and irrelevant information, to the latent representation z characterized by a low-dimensional distribution of effective features, via the application of nonlinear activation function g, as follows in Equation (10).
(10)z=g(Wx+b)

Subsequently, the decoder performs the inverse operation of the encoder. The latent representation z is processed to reproduce the input signal via activation function f, defined as Equation (11).
(11)$$x^{\prime }=f(W^{\prime }z+b^{\prime })$$
where W, b and W′, b′ denote the weight and bias parameters of the encoder and decoder, respectively. These values are iteratively updated through backpropagation to minimize the loss value between the desired input x and output x′. Mean square error (MSE) is defined as the loss function in Equation (12).
(12)$$loss={\left\|x-x^{\prime }\right\|}^{2}$$

Within the optimization process of network parameters, we obtained the latent representation z by minimizing the value of MSE. Then, z, containing more meaningful information from the input date x, is for further non-CAD and CAD classification.

### 2.5. Statistical Analysis

Statistical analysis is crucial to evaluate the effectiveness of multi-modal features. We verify the normal distribution of features using the Kolmogorov–Smirnov test, and these features from different groups are analyzed using Student’s *t*-test, while features with non-normal distribution are evaluated using Mann–Whitney U test. *p*-value < 0.05 denotes statistical difference between features from different groups.

### 2.6. Classification and Evaluation

After extracting and compressing deep-coding features, this study employed a recursive feature elimination (RFE) algorithm to evaluate the contribution rate of each feature and rank them for further classification tasks [45]. RFE is a simple and robust algorithm in processing small-sample data. It iteratively removes the feature with the least important score until the optimal features are selected.

The selected optimal feature subset was fed into the SVM classifier for identifying CAD. The combination of deep-coding features and a conventional classifier effectively mitigates over-fitting in small-sample datasets, achieving higher classification accuracy with reduced training parameters and accelerated processing speed. Consequently, deep-coding features were sent to the SVM classifier based on both linear kernel and radial basis function (RBF) kernel to identify CAD in this study. The linear kernel employs a hyperparameter C, and the radial basis function kernel utilizes two hyperparameters C and r. These optimal hyperparameter values are trained through grid search with specified ranges; the range of C is 2*^n^* and *n* is the integer from −4 to 13, and the range of r is 2*^m^* and *m* is the integer from −7 to 6. These address potential over-fitting and the classifier’s nonlinear behavior. To validate the model identification performance, the 5-fold cross-validation method is performed, ensuring that training and testing samples are sourced from distinct subjects to uphold the reliability of results.

Four widely accepted evaluation metrics for anomaly classification are used to assess classification performance, including accuracy (*ACC*), sensitivity (*SEN*), specificity (*SPE*) and f1-score (*F*1).
(13)$$ACC=\frac{tp+tn}{tp+fp+tn+fn}\times 100\%$$
(14)$$SEN=\frac{tp}{tp+fn}\times 100\%$$
(15)$$SPE=\frac{tn}{tn+fp}\times 100\%$$
(16)$$F1=\frac{2tp}{2tp+fp+fn}\times 100\%$$
where *tp* and *tn* are the number of positive and negative samples correctly identified, respectively. And *fp* and *fn* are the number of positive and negative samples incorrectly identified, respectively.

## 3. Results

This study executed signal pre-processing, ECG-PCG coupling signal evaluation, MRP construction and statistical analysis in MATLAB R2020b. Deep learning network and classification were conducted in Python 3.9. This section illustrates all results of the proposed method.

### 3.1. Comparison of Single- and Multi-Modal Data

This study leverages single ECG, PCG and ECG-PCG coupling features to form multi-modal information for better classifying non-CAD and CAD cases. The reason is that these single-modal signals have a close correlation with CAD. Specifically, ECG and PCG are derived from different cardiovascular activities, while ECG-PCG coupling information reflects the inherent correlation between these activities. In order to obtain more adequate multi-modal representations, each CNN branch in the parallel CNN network encodes deep MRP features from each single-modal signal. Subsequently, the extracted deep-coding MRP features are further compressed as 1 × 400 dimension space by the autoencoder network. Following data compression, three single-modal features are integrated into multi-modal information for CAD classification.

Figure 7 depicts the average classification accuracy trend during the five-fold cross-validation using single- and multi-modal data. With an increasing number of more meaningful features, average accuracy rises sharply. When the accuracy reaches the optimal value, it begins to decline with an increasing number of features. This indicates that the high-dimensional features contain redundant information, resulting in an adverse classification effect. Table 4 details the classification results of different modal data. Compared with the results of all single-modal signals, multi-modal information integrating ECG, PCG and ECG-PCG coupling features acquires superior detection performance, with accuracy, sensitivity, specificity, and F1-score of 98.49%, 98.57%, 98.57%, and 98.89%, respectively. Among single-modal signals, the classification performance of ECG-PCG coupling features is superior to single ECG or single PCG. The reason is that the ECG-PCG coupling signal based on deconvolution of ECG and PCG is attributed to the additional novel coupling information, which transcends information about ECG and PCG.

### 3.2. Overall Classification Results of Multi-Modal Method

To assess comprehensively the performance of our proposed multi-modal learning method, a five-fold cross-validation strategy is adopted. Table 5 shows each fold and average results in five-fold cross-validation based on the stratified sampling principle. With fusing different single-modal features to form multi-modal information, our proposed method achieves remarkable results, with one-fold and two-fold validations of the multi-modal method yielding the highest accuracies of 100%. Across all five-fold validation results, the mean accuracy remains consistently high, at 98.49%, accompanied by a modest standard deviation of 1.24%, underscoring the robustness and stability of our proposed method.

### 3.3. Features Analysis of Different Modal Signals

In the classification framework of our multi-modal learning method, RFE embedded in SVM selects more salient features based on the classification contribution rates in different modal signals. Following feature compression based on the autoencoder network, each single-modal signal gains 400 features for analyzing non-CAD and CAD cases. Within an increasing number of significant features, the classification accuracy rate rises quickly. However, due to the presence of the inherent information redundancy, the accuracy rate begins to decrease when the optimal feature count is surpassed, with the highest accuracy in each single-modal signal.

To analyze the importance of each feature in CAD detection, we employ statistical analysis to assess the difference between features of non-CAD and CAD groups. As shown in Figure 8, most ECG deep-coding features with *p*-values ≥ 0.05 show no statistical difference. In contrast, the ECG-PCG coupling signal has the most features with a significant difference. It indicates that the ECG-PCG coupling signal provides the highest contribution rate and effective information for CAD detection, whereas the ECG signal offers the least information in the classification task. These results are consistent with Figure 7. Due to the existence of numerous features with no statistical difference, we select optimal features by RFE for achieving the best detection result.

As seen from Figure 7 and Figure 8, not all features with significant differences are helpful in detecting CAD. Thus, we also analyze the correlation between all features. Correlation coefficients between two features in multi-modal features are computed and are painted as heat maps, as shown in Figure 9. We can see that single-modal features have high correlation coefficients, indicating that only a few features can characterize all signal information. Correlation coefficients between different single-modal features are small, suggesting that there is low correlation between different single-modal features. Different single-modal features comprise positive or negative information for detection CAD. Consequently, by combining different single-modal features, a finite number of optimal features selected can improve the classification result.

Additionally, the top five optimal features based on the contribution rates of each single-modal signal are further analyzed for their efficiency, as shown in Figure 10. ECG features contain f-ecg1~f-ecg5, PCG features span f-pcg1~f-pcg5, and ECG-PCG coupling features include f-e-pcg1~f-e-pcg5. By analyzing the statistical significance (*p*-value < 0.05) between each feature and the class label, we observe the clearest differences between non-CAD and CAD subjects in the ECG-PCG coupling signal. This indicates the superior classification contributions of ECG-PCG coupling features compared to ECG or PCG features. Furthermore, the combination of different single-modal information sources, with each single-modal signal features carrying complementary information, fosters a notable improvement of classification in the multi-modal learning method.

### 3.4. Performance Analysis of Different Models

The complexity of the deep learning network and feature reduction may affect the classification accuracy. We remove the autoencoder model from our model and achieve a reasonable result with accuracy, sensitivity, specificity, and f1-score of 98.50 ± 1.22, 100.00 ± 0.00, 98.67 ± 2.16 and 98.91 ± 0.89, but it lost more time.

To confirm the superiority of the proposed method, we compare our method with other advanced models, including ResNet50-based and transformer-based models, and obtain the classification results, as shown in Table 6. We initially use the ResNet50 structure to replace each 2-D CNN branch in our method. ResNet50 contains residual modules, which can enhance more effective information and reduce information loss. The ResNet50-based model yields good accuracy of 90.96%, which is lower than our method, indicating that deeper networks can encode less meaningful features. Similarly, we also build a transformer-based model to validate its performance. By using the transformer structure to replace each 2-D CNN branch, multi-modal signals are fed into the new model to encode deep features based on the advantages of transformer in 1-D signals. However, the transformer-based model achieves lower accuracy of 88.46%. The reason may be that the small-sample dataset limits the model performance.

In addition, the strong classifier is key to improving detection accuracy. Our study employs five traditional classifiers to identify non-CAD and CAD subjects and yields detection results, as shown in Table 7. The SVM classifier achieves the best classification accuracy.

## 4. Comparison and Discussion

To overcome information shortages of single-modal data, this study proposes a novel multi-modal learning method to integrate ECG, PCG and ECG-PCG coupling features for CAD detection. The proposed model firstly conducts deconvolution of ECG and PCG to produce an ECG-PCG coupling signal, which acquires the inherent related information of cardiovascular activities. And then, we introduce the MRP to quantify the nonlinear dynamic characteristics of each single signal. The network integrating deep learning is designed to encode deep spatial features of each MRP, and fuse them for multi-modal feature extraction, data compression and classification. By leveraging the complementary strengths of multi-modal data and the advantage of deep learning, we address the limitations of the single-modal method and attain remarkable improvements in CAD detection.

As previously mentioned, the ECG signal undergoes significant alterations, particularly in the case of advanced CAD, manifested through various changes such as ST-wave elevation or depression and T-wave inversion. These changes are markedly distinguishable between non-CAD and CAD subjects [4]. Similarly, when coronary arteries occlude to a certain degree, blood flows through the narrowed coronary artery and forms turbulence. Then, weak murmurs begin to occur and PCG waveform change appears gradually [5,6]. Consequently, more significant differences in PCG between non-CAD and CAD appear. However, when CAD patients show more than 90% blockage, one or more coronary arteries are mostly blocked. It leads to a reduction in blood flow and subsequent disappearance of murmurs. Hence, this complexity poses a challenge in accurately distinguishing CAD from non-CAD. Additionally, coronary abnormality disrupts the inherent harmony between cardiac electrical and mechanical activities, prompting us to develop a novel ECG-PCG coupling signal to reflect these changes based on deconvolution of ECG and PCG. To assess the nonlinear characteristic distribution of each single signal between non-CAD and CAD subjects, this study introduces MRP to visually display their differences, as shown in Figure 2 and Figure 3.

In the context of CAD analysis utilizing single-modal data, ECG carries important information of cardiac electrical activity, which easily distinguishes severe CAD patients from non-CAD subjects due to significant ECG waveform alterations in the terminal stage. Nevertheless, the subtle ECG waveform change poses challenges in classifying moderate CAD patients. Conversely, PCG reflects the mechanical activity of the heart and it shows notable waveform change in moderate CAD patients. However, detecting severe CAD patients with little change in PCG signal becomes complicated due to the disappearance of murmurs. As a result, single ECG or single PCG only provides single-aspect information for CAD detection, and leads to lower classification performance, as shown in Figure 7 and Table 4. To enhance detection ability, we innovatively produce the novel ECG-PCG coupling signal by conducting deconvolution of ECG and PCG. By integrating ECG, PCG and ECG-PCG coupling information, the multi-modal learning method improves classification results via deep feature encoding. This underscores the presence of complementary information within single-modal data and demonstrates that identification ability of multi-modal data outperforms that of single-modal data. Notably, ECG-PCG coupling features yield higher results among all single-modal signals, with accuracy of 84.94%, and multi-modal information obtains the highest accuracy of 98.49%. This indicates that the proposed multi-modal learning method effectively overcomes information shortages in single-modal data.

Nonlinear analysis proves more adept at capturing the dynamic characteristics of the cardiac signal for its inherent nonlinear nature. Thus, this proposed model introduces an MRP to quantify each single-modal signal. To enrich disease-related information, we leverage the advantages of the deep learning network by constructing the parallel CNN network. This network encodes ECG, PCG and ECG-PCG coupling MRPs and subsequently concatenates all single-modal deep-coding features. Considering the existence of information redundancy in the extracted high-dimensional features, an autoencoder network following behind the parallel CNN network is added, facilitating feature compression. Ultimately, each single-modal signal obtains 400 features and then combine all single-modal features to form multi-modal information for CAD detection. Feature analysis indicates that not all features are helpful in identifying CAD, as shown in Figure 8, Figure 9 and Figure 10. To select the most discriminative features, an embedded RFE algorithm is employed to estimate the classification importance score of each feature, and ranks them to select optimal features. ECG-PCG coupling features obtain superior results to ECG or PCG in single-modal analysis, while multi-modal data achieve the best classification results.

Table 8 summarizes previous studies on CAD detection based on single ECG or single PCG or both signals, alongside their corresponding testing results on the public or self-collected data. Kaveh et al. [22] used ECG multiple domain features for CAD classification, and validated their model on the MIT-BIH ECG database. In the works of Li et al. [20], Samanta et al. [21] and Huang et al. [46], CAD was analyzed by extracting different PCG features utilizing machine or deep learning methods. Notably, Li et al. [47] combined ECG and PCG for CAD detection by extracting multi-domain deep features. To ensure a fair comparison with existing studies, we further validated our proposed model on the dataset previously used by Li et al. [47], and achieved comparable classification performance, with accuracy, sensitivity and specificity of 96.37%, 98.26% and 90.22%, respectively. However, considering the challenge of imbalanced data, which can restrict the generalization ability of the model, this study solved this issue by randomly selecting 60 patients from all CAD patients to eliminate the imbalanced condition, and then retrained the classification model on the new balanced dataset. The new model, trained on the new balanced data, achieved remarkable detection performance, with accuracy, sensitivity and specificity of 97.08%, 98.12% and 96.22% in classifying 60 non-CAD subjects and 60 CAD patients. Additionally, we also augmented the data quantity by segmenting the ECG and PCG signals of non-CAD subjects into 10 s segments with an overlap of 50% in our study, so that the ratio of non-CAD and CAD sample size was approximately balanced (64 × 60:135 × 30). The augmentation dataset obtained stable performance, with accuracy, sensitivity and specificity of 98.10%, 98.43% and 97.78%, respectively. Furthermore, each signal segment cost an average of 13.15 s, which indicates that this proposed method is a feasible and real-time approach in clinical practice.

In the realm of CAD detection based on single ECG or single PCG classification, numerous previous methods have been validated on two widely used open-source public databases, namely the PhysioNet ECG database and the PhysioNet/CinC Challenge 2016 PCG database. To fairly compare with previous methods, the proposed multi-modal learning method was also validated on these two public databases, and all details are listed in Table 9. In CAD detection using a public ECG database, Kumar et al. [8] and Acharya et al. [10] relied on hand-crafted features obtained by a machine learning method, whereas Tan et al. [9] adopted time-domain encoding features derived from 1D-CNN. More recently, the potential of PCG signals for CAD detection has been addressed, with most authors using hand-crafted or deep-coding features [11,13]. Notably, Noman et al. [12] employed MFCC deep features encoded by 2D-CNN, while Humayun et al. [14] constructed 1D-CNN to extract PCG temporal–spatial features for classification. Due to there being only single-modal signals in these two public databases, our model encoded single-modal features from both ECG and PCG databases for identifying CAD, and yielded better results, with accuracy of 99.87% and 97.56%, respectively. This underscores the feasibility and robust generalization capabilities of this proposed model in CAD detection.

It should be noted that this study still has some limitations. During the data collection, we only record single ECG and single PCG, overlooking the potential benefits of incorporating multiple modalities. The inclusion of additional data sources, such as imaging techniques or other data, could significantly enrich identifying information and improve the classification capabilities. Moreover, the imbalance distribution of data points across classes poses a challenge in achieving optimal classification performance. Balanced data is crucial for obtaining more reliable results. Consequently, data augmentation strategies and the acquisition of additional data are essential for validating and refining our proposed model. In future, we will incorporate additional modal data to capture a broader spectrum of information for addressing information limitations. This will enable us to perform a more comprehensive analysis and validation of the proposed method. Furthermore, we plan to expand our data with a wider variety of samples to mitigate class imbalance and further enhance the detection performance of our model. Ultimately, our goal is to develop a robust and comprehensive CAD detection system that can provide more meaningful insights to patients and healthcare professionals.

## 5. Conclusions

Single-modal data inherently suffer from information shortages, whereas multi-modal information is pivotal in enhancing the performance of CAD detection. This study proposes a multi-modal learning method to integrate ECG and PCG for the purpose of detecting CAD. Besides ECG and PCG, we successfully produce a novel ECG-PCG coupling signal by conducting deconvolution of ECG and PCG. Considering the inherent nonlinear and non-stationary nature of physiological signals, we construct the MRP to quantify ECG, PCG and ECG-PCG coupling signals, respectively. A network integrating deep learning by incorporating the parallel CNN network and the autoencoder network is designed for multi-modal deep feature encoding, fusion and compression. Subsequently, optimal features are selected based on the classification contribution rate of each feature, which is then sent to the SVM classifier for final classification between non-CAD and CAD cases. Classification results conclusively demonstrate that multi-modal information, encompassing all single-modal features, achieves an improvement of detection performance compared with single-modal data. The multi-modal learning method is a feasible and robust technique for CAD detection and diagnosis.

## Figures and Tables

**Figure 1 sensors-24-06939-f001:**
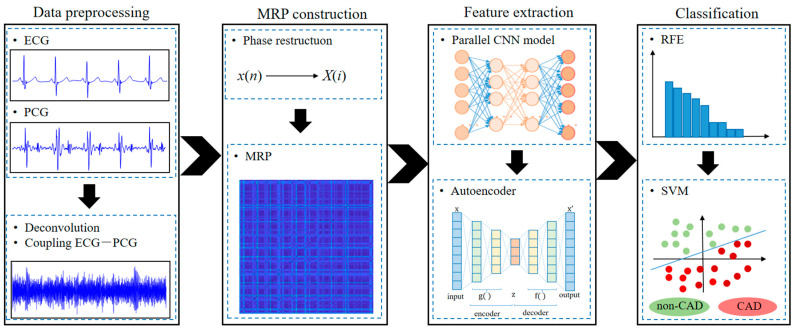
The block diagram of this proposed method, including data preprocessing, MRP construction, feature extraction and classification sections.

**Figure 2 sensors-24-06939-f002:**
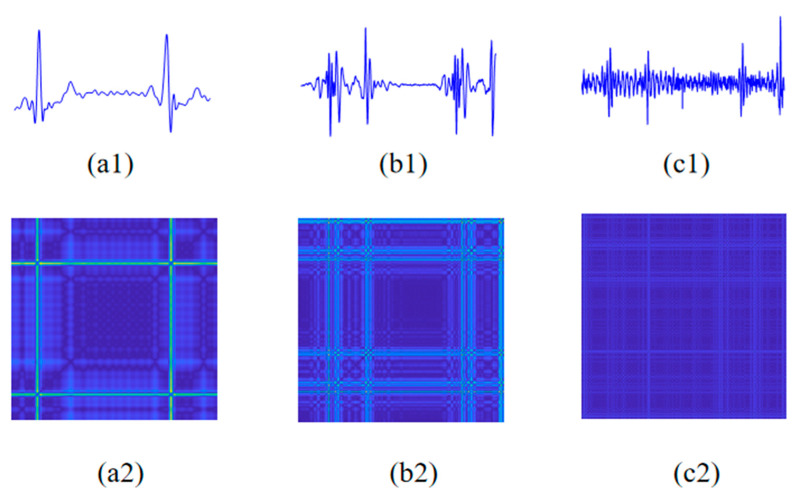
ECG, PCG and ECG-PCG coupling signals and MRPs of a non-CAD subject. (**a1**) ECG signal. (**a2**) MRP of ECG signal. (**b1**) PCG signal. (**b2**) MRP of PCG signal. (**c1**) ECG-PCG coupling signal. (**c2**) MRP of ECG-PCG coupling signal. Note: the brighter points represent farther distances.

**Figure 3 sensors-24-06939-f003:**
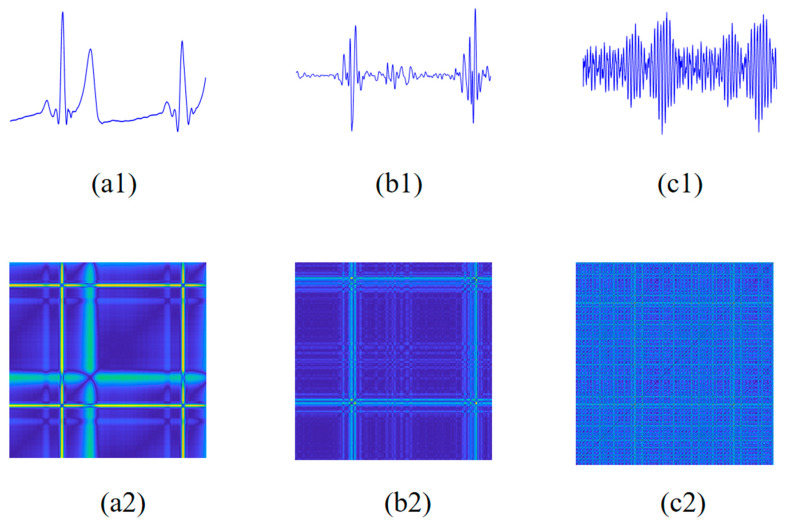
ECG, PCG and ECG-PCG coupling signals and MRPs of a CAD patient. (**a1**) ECG signal. (**a2**) MRP of ECG signal. (**b1**) PCG signal. (**b2**) MRP of PCG signal. (**c1**) ECG-PCG coupling signal. (**c2**) MRP of ECG-PCG coupling signal. Note: the brighter points represent farther distances.

**Figure 4 sensors-24-06939-f004:**
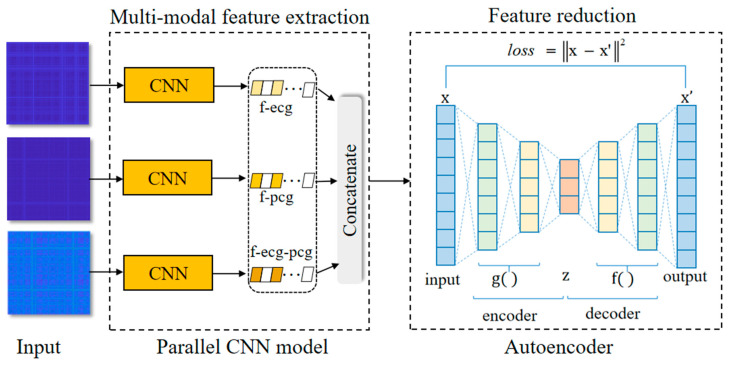
Frame of the integrating deep learning network. Parallel CNN model encodes deep features from its input MRPs and autoencoder reduces the concatenated multi-modal features.

**Figure 5 sensors-24-06939-f005:**
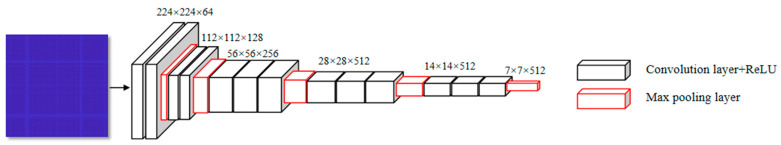
The structure of each CNN in the parallel CNN network. CNN includes multiple convolutional and max pooling layers encoding deep features.

**Figure 6 sensors-24-06939-f006:**
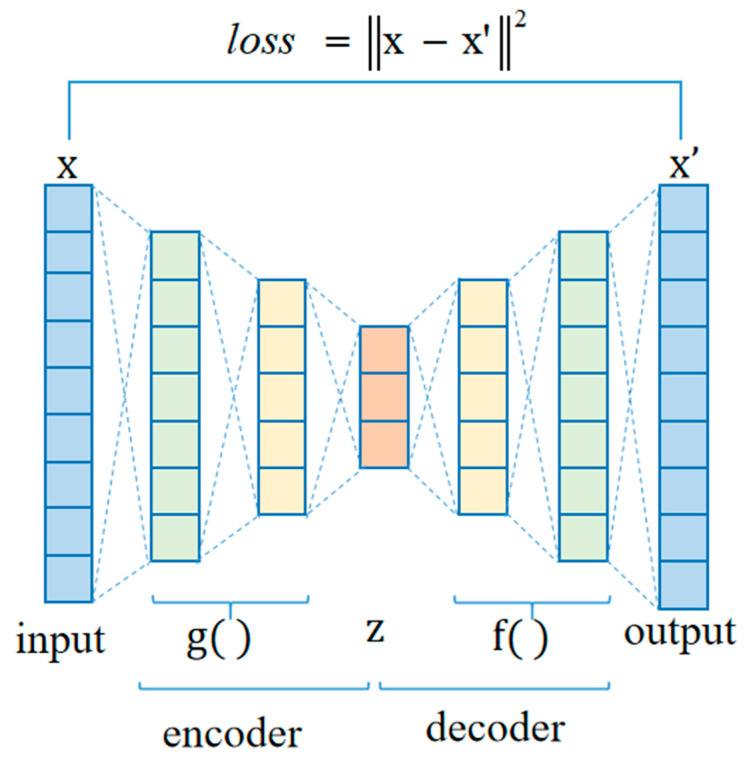
Structure of the autoencoder network. x is the input feature vector and x′ is the output reconstructed feature vector. z is the latent representation encoded as the reduced feature vector.

**Figure 7 sensors-24-06939-f007:**
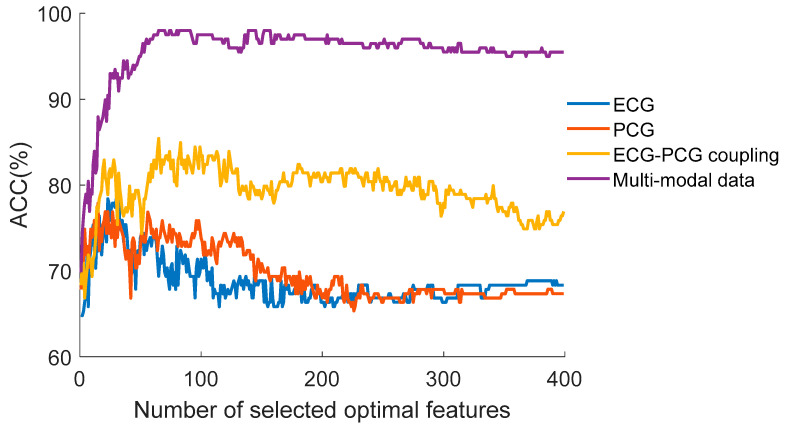
The average accuracy trend of single- and multi-modal data with increasing features.

**Figure 8 sensors-24-06939-f008:**
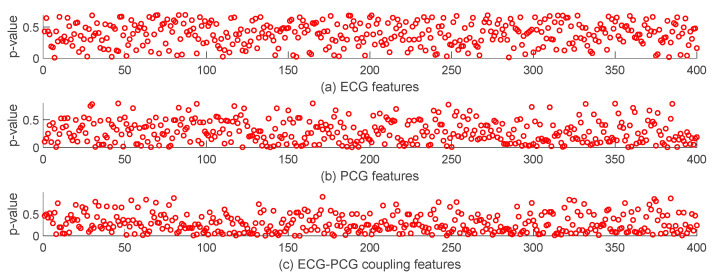
The *p*-value of each single-modal feature between non-CAD and CAD. (**a**) The *p*-value of ECG features. (**b**) The *p*-value of PCG features. (**c**) The *p*-value of ECG-PCG coupling features.

**Figure 9 sensors-24-06939-f009:**
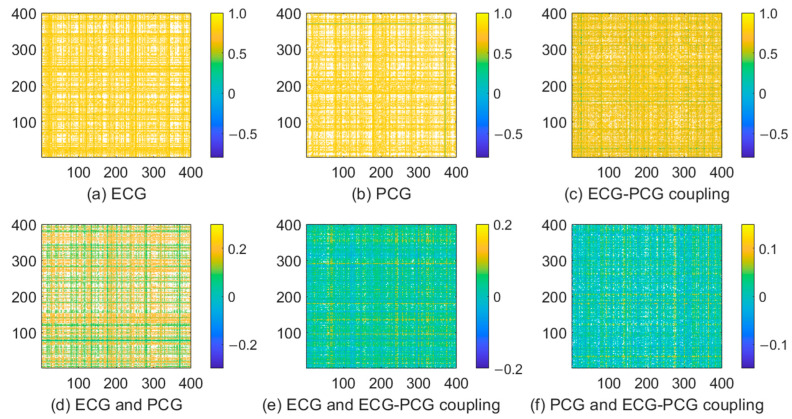
Heat maps of correlation coefficients between features of different signals. (**a**) Correlation coefficients between ECG features. (**b**) Correlation coefficients between PCG features. (**c**) Correlation coefficients between ECG-PCG coupling features. (**d**) Correlation coefficients between ECG and PCG features. (**e**) Correlation coefficients between ECG and ECG-PCG coupling features. (**f**) Correlation coefficients between PCG and ECG-PCG coupling features.

**Figure 10 sensors-24-06939-f010:**
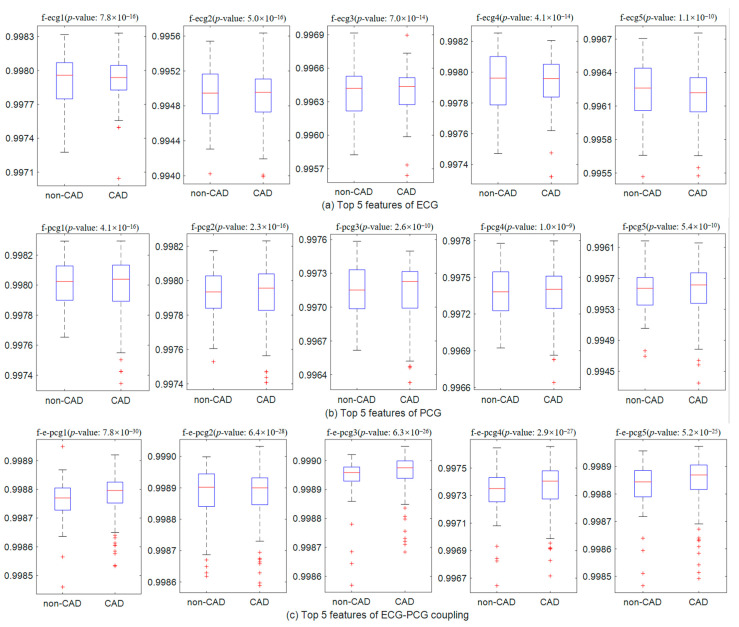
Significant ECG, PCG and ECG-PCG coupling features analyzed by the proposed multi-modal learning method. The top 5 most significant features of each single-modal signal are selected and the caption contains their names and *p*-values. (**a**) Top 5 features of ECG signal. (**b**) Top 5 features of PCG signal. (**c**) Top 5 features of ECG-PCG coupling signal.

**Table 1 sensors-24-06939-t001:** Basic information of all subjects (mean ± SD).

Characteristics	Non-CAD	CAD
Age	61 ± 10	62 ± 10
Male/female	30/34	89/46
Height	164 ± 7	166 ± 8
Weight	69 ± 12	71 ± 11
Heart rate	72 ± 12	75 ± 16
Systolic blood pressure	134 ± 15	133 ± 16
Diastolic blood pressure	80 ± 11	82 ± 12

**Table 2 sensors-24-06939-t002:** All parameters of each CNN in the parallel CNN model.

Index	Layer	Index	Layer
1	conv3_64	10	max-pooling_2
2	conv3_64	11	conv3_512
3	max-pooling_2	12	conv3_512
4	conv3_128	13	conv3_512
5	conv3_128	14	max-pooling_2
6	max-pooling_2	15	conv3_512
7	conv3_256	16	conv3_512
8	conv3_256	17	conv3_512
9	conv3_256	18	max-pooling_2

Note: “conv (kernel size)_(number of kernels)” represents the convolutional parameters, and “max-pooling_(kernel size)” represents the max-pooling parameters.

**Table 3 sensors-24-06939-t003:** All parameters of the proposed autoencoder network.

Indicator	Parameter	Indicator	Parameter
Structure	2000-1000-400-1000-2000	Learning rate	0.001
Optimizer	SGD	Batch	32
Loss	MSE	Epoch	1000

**Table 4 sensors-24-06939-t004:** The detailed classification results of single- and multi-modal data.

Modal Signal	*ACC* (%)	*SEN* (%)	*SPE* (%)	*F*1 (%)
ECG	79.38 ± 4.36	92.59 ± 4.68	51.54 ± 5.76	61.75 ± 7.56
PCG	77.88 ± 1.92	91.85 ± 4.32	48.21 ± 11.63	57.54 ± 7.72
ECG-PCG coupling	84.94 ± 4.97	94.81 ± 6.87	64.10 ± 3.24	73.67 ± 6.12
Multi-modal data	98.49 ± 1.24	98.57 ± 1.75	98.57 ± 2.86	98.89 ± 0.90

**Table 5 sensors-24-06939-t005:** Each fold and average results of multi-modal method using five-fold cross-validation.

Number-Fold	*ACC* (%)	*SEN* (%)	*SPE* (%)	*F*1 (%)
1-fold	100.00	100.00	100.00	100.00
2-fold	100.00	100.00	100.00	100.00
3-fold	97.50	96.43	100.00	98.18
4-fold	97.50	100.00	92.86	98.11
5-fold	97.44	96.43	100.00	98.18
mean ± std	98.49 ± 1.24	98.57 ± 1.75	98.57 ± 2.86	98.89 ± 0.90

**Table 6 sensors-24-06939-t006:** Classification results of different deep learning-based models.

Model	*ACC* (%)	*SEN* (%)	*SPE* (%)	*F*1 (%)
ResNet50-based model	90.96 ± 2.89	94.81 ± 1.81	82.82 ± 5.71	85.45 ± 4.80
Transformer-based model	88.46 ± 3.35	93.33 ± 2.77	78.21 ± 8.81	81.21 ± 5.61
Our model	98.49 ± 1.24	98.57 ± 1.75	98.57 ± 2.86	98.89 ± 0.90

**Table 7 sensors-24-06939-t007:** Classification results of different classifiers.

Classifier	*ACC* (%)
Decision tree	88.34
Linear Discriminant Analysis	81.65
Bayse	81.36
KNN	90.83
SVM	98.49

**Table 8 sensors-24-06939-t008:** Summary of existing studies on the diagnosis of CAD.

Author	Data	Method	Result (%)
Li et al. [20]	Self-collected 135 CAD/60 non-CAD	PCG, multi-domain features, deep features, MLP	*ACC*: 90.4 *SPE*: 83.4 *SEN*: 93.7
Samanta et al. [21]	Self-collected 29 CAD/37 non-CAD	PCG, time domain and frequency domain features, CNN	*ACC*: 82.6 *SPE*: 79.6 *SEN*: 85.6
Kaveh et al. [22]	MIT-BIH 43 CAD/46 non-CAD	ECG, time domain and frequency domain features, SVM	*ACC*: 88.0 *SPE*: 92.6 *SEN*: 84.2
Huang et al. [46]	Self-collected 348 Normal/206 CAD	PCG, MFCCs, PCG sequence, Customized model	*ACC*: 96.05 *SPE*: 96.12 *SEN*: 96.12
Li et al. [47]	Self-collected 347 CAD/74 non-CAD	ECG and PCG, sequence, spectrum image, ST image, MFCCs image	*ACC*: 96.51 *SPE*: 90.08 *SEN*: 99.37
This study	Self-collected 135 CAD/64 non-CAD	ECG and PPG, Multi-modal deep-coding features, SVM	*ACC*: 98.49 *SPE*: 98.57 *SEN*: 98.57
	Self-collected [39] 135 CAD/60 non-CAD	ECG and PCG, Multi-modal deep-coding features, SVM	*ACC*: 96.37 *SPE*: 90.22 *SEN*: 98.26
	Self-collected 60 CAD/60 non-CAD	ECG and PCG, Multi-modal deep-coding features, SVM	*ACC*: 97.08 *SPE*: 96.12 *SEN*: 98.22

**Table 9 sensors-24-06939-t009:** Comparison of existing studies on ECG classification using the PhysioNet dataset and PCG classification using the PhysioNet/CinC Challenge 2016 dataset.

Author	Classification Method	Input	Result (%)
Studies on ECG classification using the PhysioNet dataset			
Kumar et al. [8]	SVM	Time–frequency features	*ACC*: 99.60
Tan et al. [9]	1-D CNN	ECG signal	*ACC*: 99.85
Acharya et al. [10]	1-D CNN	Entropy features	*ACC*: 99.27
This study	SVM	MRP deep-coding features	*ACC*: 99.87
Studies on PCG classification using the PhysioNet/CinC Challenge 2016 dataset			
Tschannen et al. [11]	1-D CNN	Time features, Frequency features	*ACC*: 87.00
Noman et al. [12]	2-D CNN	MFCCs image	*ACC*: 88.80
Baydoun et al. [13]	Boosting and bagging model	Time–frequency features, Statistical features	*ACC*: 91.50
Humayun et al. [14]	1D-CNN	PCG signal	*ACC*: 97.50
This study	SVM	MRP deep-coding features	*ACC*: 97.56

## Data Availability

The data are available on request.

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
