# Peer review of "Coronary Artery Disease Detection Based on a Novel Multi-Modal Deep-Coding Method Using ECG and PCG Signals"

_sensors, 2024, doi:10.3390/s24216939_

Round 1

Reviewer 1 Report

Comments and Suggestions for Authors

Major issues:

1.      When applied to larger or more diversified datasets, the study may be less effective, and autoencoding feature dimensions may affect diagnosis accuracy. I expect convincing explanation in this regard.

2.      The model is accurate, however it is not compared or integrated with more advanced models like Transformer-based models, which could improve it. Other classifiers were not evaluated, however multi-modal features and SVM classifiers were accurate.

3.      Deep learning and autoencoder networks for feature dimension reduction may lose important information, decreasing CAD detection accuracy.

4.      The research improves multi-modal learning but faces information loss during feature detection, model complexity, and dimensionality reduction. How to avoid this situation?

5.      The model's complexity is raised by using a parallel CNN and autoencoder, needing greater processing power and time for implementation.

6.      The diagnosis of coronary artery disease (CAD) associated with low or moderate blockage may be limited to patients with ≥50% blockage.

7.      Nonlinear dynamic analysis may be used to examine physiological data, however the complexity of these signals can still lead to misdiagnosis in specific circumstances.

8.      It need explaining recursive feature elimination (RFE)'s flowchart and details

9.      The study addresses overfitting in small-sample datasets, but its low sample size may hinder generalization due to less robust model training and assessment.

10.  Hyperparameters like C and rare crucial for a model's usefulness, but grid search may not be the most robust strategy, potentially limiting the model's effectiveness.

11.  The mean heart rate (72±12 vs. 75±16) may be somewhat higher in CAD, but is just suggestive and not sufficient to definitely diagnose CAD.

12.  The accuracy of data analysis increases with the number of data points, but it decreases after a certain point due to excess adverse categorization effects, requiring further investigation.

Minor issues:

1.      Although the RFE algorithm is used, the feature selection process may require more advanced analysis or feature prioritization.

2.      The study did not include other data sources, such as imaging technology, that could have helped enrich the diagnostic process.

3.      Augmentation techniques are important to improve the diversity and quantity of data, hence boosting the validity and effectiveness of the classification model.

4.      The model makes no reference to online or real-time training, which may be necessary for clinical use.

5.      The linear and non-linear behavior of the ECG-PCG coupling signal is not clearly stated.

Reviewer 2 Report

Comments and Suggestions for Authors

1. The authors used inconsistent definitions of CAD in the context of the methods sections in Line 153(those with blockages ≥50% in at least one major coronary artery)  and in the discussion sections in Line 458(within CAD patients with more than 20% coronary obstruction). 

2.  How to differentiate myocardial ischemia in different coronary branch sites using a single II lead electrocardiogram?

3.  In this manuscript, the authors did not indicate the collection location of the phonocardiogram signal.

4. In Line 463, “when CAD patients show more than 90% blockage, coronary arteries are mostly blocked”.  Does “90% blockage” refer to the blockage of all three main branches of the coronary artery? Or only a blockage in one of the branches?  Is there any difference between segmental stenosis and diffused stenosis?

5.  The phonocardiogram is a mixed waveform of filtered chest sounds, and the effect of a single coronary artery branch stenosis on the murmurs in the phonocardiogram is unclear. In Line 463,  The authors'mentioned that "coronary arteries are mostly blocked. It leads to reduction of blood flow and subsequent disappearance of murmurs."  This explanation is not sufficient and convincing. Please provide a detailed interpret to the pathophysiological mechanisms that lead to the disappearance of murmurs.

Comments on the Quality of English Language

.

Reviewer 3 Report

Comments and Suggestions for Authors

comment in the attached file

Round 2

Reviewer 1 Report

Comments and Suggestions for Authors

The manuscript has been improved a lot to recommend it for publicationn in Sensors.